# Estimation and Sharpening of Blur in Degraded Images Captured by a Camera on a Moving Object

**DOI:** 10.3390/s22041635

**Published:** 2022-02-19

**Authors:** Toshiyuki Hayashi, Takashi Tsubouchi

**Affiliations:** 1Graduate School of System and Information Engineering, University of Tsukuba, Tsukuba 305-8573, Japan; 2Faculty of System and Information Engineering, University of Tsukuba, Tsukuba 305-8573, Japan; tsubouchi.takashi.gw@u.tsukuba.ac.jp

**Keywords:** Point Spread Function (PSF), image sharpening method, cepstrum analysis, estimation of motion blur, feature point of speckled texture

## Abstract

In this research, we aim to propose an image sharpening method to make it easy to identify concrete cracks from blurred images captured by a moving camera. This study is expected to help realize social infrastructure maintenance using a wide range of robotic technologies, and to solve the future labor shortage and shortage of engineers. In this paper, a method to estimate parameters of motion blur for Point Spread Function (PSF) is mainly discussed, where we assume that there are two main degradation factors caused by the camera, out-of-focus blur and motion blur. A major contribution of this paper is that the parameters can properly be estimated from a sub-image of the object under inspection if the sub-image contains uniform speckled texture. Here, the cepstrum of the sub-image is fully utilized. Then, a filter convoluted PSF which consists of convolution with PSF (motion blur) and PSF (out-of focus blur) can be utilized for deconvolution of the blurred image for sharpening with significant effect. PSF (out-of-focus blur) is a constant function unique to each camera and lens, and can be confirmed before or after shooting. PSF (motion blur), on the other hand, needs to be estimated on a case-by-case basis since the amount and direction of camera movement varies depending on the time of shooting. Previous research papers have sometimes encountered difficulties in estimating the parameters of motion blur because of the emphasis on generality. In this paper, the main object is made of concrete, and on the surface of it there are speckled textures. We hypothesized that we can narrow down the candidates of parameters of motion blur by using these speckled patterns. To verify this hypothesis, we conducted experiments to confirm and examine the following two points using a general-purpose camera used in actual bridge inspections: 1. Influence on the cepstrum when the isolated point-like texture unique to concrete structures is used as a feature point. 2. Selection method of multiple images to narrow down the candidate minima of the cepstrum. It is novel that the parameters of motion blur can be well estimated by using the unique speckled pattern on the surface of the object.

## 1. Introduction

In Japan, there are more than 700,000 bridges managed by the national and local governments [1].The percentage of bridges that are 50 years old from date of construction will increase to more than 50 percent in 2029 and it is required by law to inspect these bridges once every five years [2]. It is difficult to secure the engineers necessary for these inspections; the number of engineers is insufficient for bridge inspection in 20 percent of local governments in Japan. Thus it is desired to realize bridge inspection using a camera mounted on a mobile object such as a drone or a mobile robot in order to solve the future labor shortage and shortage of engineers and improve the efficiency of inspections [3]. While the elemental technologies and methods for movement and inspection differ depending on the type of robotic technology, the bridge inspection technology introduced by the Ministry of Land, Infrastructure, Transport and Tourism in Japan mainly uses a general-purpose camera with tens of millions of pixels to acquire images (video or still images) [4]. In order to obtain information necessary for maintenance from the images acquired by the camera, the camera can be mounted on a mobile object such as a drone or a mobile robot. On the other hand, the images captured by the camera have the following problems: The camera tends to be out of focus because the images are taken in a dark place; the performance of the camera is general-purpose, there is a limitation in narrowing down the shutter speed in order to obtain the amount of light necessary for taking pictures because some pictures are taken in dark place; or the acquired images are blurred because images are captured while moving on the mobile platform. The blurred image (hereinafter represented by “degraded image”) makes it difficult to check for cracks or other damages necessary for social infrastructure maintenance [5]. The motivation of this research is to realize bridge inspection managed by the national and local governments using a general-purpose camera mounted on a mobile object such as a drone or a mobile robot.

The target of our research is that degraded images, captured by a general-purpose camera mounted on a mobile object, can be sharpened by image processing using a general-purpose PC, so that the cracks in the concrete can be easily and accurately confirmed for bridge inspection. For the detection of cracks in concrete structures, there have been studies on sharpening using median filter [6,7], Gray–Hough transform [8] and Gabor filter [9]. These methods are effective for blurring of focus, but their effects have not been confirmed for “blurring” caused by camera movement. There have been recent studies on bridge inspections using a camera mounted on a mobile robot [10] or UAV [11,12,13]. In order to obtain the information necessary for bridge inspection from the images acquired by UAVs or mobile robots, it is necessary to perform advanced processing on the images, which is difficult to perform on a general-purpose PC. These prior research examples do not identify the point spread function (hereinafter represented by “PSF”) as the cause of the blur in order to sharpen the blurred image.

On the other hand, we seek the cause of the blur in the PSF and try to sharpen the blur by identifying the model of the PSF. Referring to Kamimura et al. [14] and Kobayashi et al. [15], we assume that there are two main degradation factors caused by the camera: PSF of the lens optical system and blur caused by camera shake. PSF of out-of-focus blur (hereinafter represented by “PSF-OOF”), which is a blur caused by PSF of the optical system, is caused by the characteristics of the lens, the focus shift during shooting and the resolution of the image sensor, can be confirmed before or after shooting because it is a constant function unique to each camera and lens. In contrast, blur caused by camera movement during shooting, such as PSF corresponding to motion blur during shooting (hereinafter represented by “PSF-MB”), needs to be estimated on a case-by-case basis since the amount and direction of camera movement varies depending on the time of shooting.

The relationship between an image *g* degraded by blurring and an unknown original image *f* can be expressed as in Equation (Equation 1) where ∗ is the convolution and *h* is PSF.
(1)g=h∗f+n

Ignoring the noise *n*, we can obtain $f=h^{-1}*g$. Theoretically, we can recover *f* if we could find $h^{-1}$ such that $h^{-1}$ is the inverse of *h* for PSF which causes degradation for *f*. This kind of restoration for blurring due to degradation is often called sharpening. Deconvolution is the process of estimating the original image *f* from a degraded image *g* by either knowing *h* or implicitly finding the equivalent of *h* in an algorithm. There has been a lot of research on sharpening degraded images due to blurring [14,15,16,17,18,19,20,21,22,23,24,25,26]. The researches can be roughly classified into two categories: Non-blind deconvolution, in which the PSF, the degradation process, is treated as known, and blind deconvolution, in which the PSF is treated as unknown. In this paper, we treat the problem as one in which the unknown PSF-MB and the known PSF-OOF are mixed before the target image is obtained, as described in Section 2. As a typical example of research on estimating unknown PSF-MB, Yoneji et al. [17], estimated the amount and direction of movement by examining the period and direction of stripes in the Fourier transformed image of the degraded image, since h(x,y) is a sinc function when the target is a motion blur. Oyamada et al. [18] also worked on estimating the amount and direction of blur movement in the direction of the minimum value from the peak of the cepstrum by using cepstrum analysis. On the other hand, in PSF-MB estimation using Fourier transform and cepstrum analysis, it may be difficult to specify the period and direction of the filtering function h(x,y) due to the influence of the original image f(x,y), which is F(u,v) after Fourier transform.

PSF-MB estimation using Fourier transform and cepstrum analysis is sometimes affected by the original image f(x,y), which makes it difficult to specify the period and direction of the filtering function h(x,y).

So the first thing we noticed was as follows. The blur that occurs in an image due to camera movement is a degradation in which a single point in the original image spreads over a certain range. In general, if the shutter speed is not too slow, the camera movement can be approximated by constant velocity linear motion. The blur can be approximated by a function that spreads linearly in width *w* only in the direction θ of the optical flow, as shown in Equation (Equation 2) [27] and this approximates PSF-MB.
(2)$$h\left(x,\;y\right)\;for\;PSF-MB=\left\{\begin{array}{cc} \frac{1}{w} & \left|x\cos\theta +y\sin\theta \right|\leq \frac{w}{2},x\sin\theta -y\cos\theta =0\\ 0 & all\;other\;cases \end{array}\right.$$

Since the cepstrum is the inverse Fourier transform of the logarithmic amplitude spectrum of the degraded image, it can be regarded as an image that emphasizes the frequency components that a certain image has strongly. The spectrum of PSF, which represents linear blur, is a sinc function with periodic zero values [19]. Given such conditions, $C_{f}$ and $C_{h}$ are the cepstrum of the original image and PSF-MB, respectively, the cepstrum $C_{g}$ of the degraded image can be expressed as $C_{g}$ = $C_{f}$ + $C_{h}$, ignoring the effect of noise. Thus, if $C_{f}$ is known, $C_{h}$ can be obtained. However, in this study, $C_{f}$ cannot be known, so it may be difficult to accurately determine the minimum value of $C_{h}$, such as when $C_{f}$ has a large value around the minimum value, simply by obtaining the cepstrum of the entire degraded image [20,21] as described in Section 2.

Therefore, when multiple candidates for $C_{h}$ are obtained in PSF-MB estimation, it is not possible to identify a single motion blur period (amount and direction of movement, hereinafter referred to as “blur amount”) that causes the motion blur, and deconvolution by PSF-MB is performed on the degraded image using each as a candidate for the blur amount. PSF-MB deconvolution must be performed on the degraded image, and the blur amount candidate estimated from the PSF-MB that shows the best sharpening effect must be identified as the blur amount [22]. For this reason, it is desirable to narrow down (specify one) $C_{h}$ candidate in PSF-MB estimation. In this paper, a method to estimate parameters which represent motion blur on the image is mainly discussed, where we assume that there are two main degradation factors caused by the camera, out-of-focus blur and motion blur. Major contribution of this paper is that the parameters can properly be estimated form a sub-image of the object under inspection if the sub-image contains uniform speckled texture. Here, cepstrum of the sub-image is fully utilized. Then, a filter which consists of convolution with the motion blur PSF and out-of-focus blur PSF can be utilized for deconvolution of the blurred image for the sharping with significant effect. Previous research papers have sometimes encountered difficulties in estimating the parameters of motion blur because of the emphasis on generality. Some studies, such as Kamimura et al. [14] and Molina et al. [23], have tackled the simultaneous estimation of two unknown “original image *f*” and “degraded kernel *h*” using Bayesian estimation, but since two unknowns are estimated at the same time, the computation Oliveira et al. [24] first estimated the direction of the blur amount, and then narrowed down the blur amount candidates by searching for pixel of minimum along that direction. Ji et al. [25] adopted a method of amplifying the $C_{h}$ to make it less sensitive to $C_{f}$. time depends on the initial settings, and the solution may diverge in some cases. These processes require time-consuming, multi-step processing on high-performance PC. On the other hand, our proposed method consists of the following processes: Extracting four small areas including feature points and performing cepstrum analysis; extracting the minimum points in eight neighborhoods using MATLAB functions for the cepstrum analysis results; and narrowing down the minimum points by calculating the average of the four results. The amount and direction of blurring can be narrowed down by simple calculations based on MATLAB functions using a general-purpose PC.

In this paper, the main object is made of concrete, and it is novel in that it is shown that the parameters of motion blur can be well estimated by using the unique speckled pattern on the surface of the object. Referring to Oyamada et al. [22] and Nohara et al. [26], we hypothesized that the influence of $C_{f}$ can be minimized by extracting proper sub-images in the originally given image and calculating the average of the cepstrum of these images, which in turn would make it possible to narrow down the minimum value candidates of the cepstrum. To verify this hypothesis, we conducted experiments to confirm and examine the following two points using a general-purpose camera [4] used in actual bridge inspections: 1. Influence on the cepstrum when the isolated point-like texture unique to concrete structures is used as a feature point. and 2. Selection method of multiple images to narrow down the candidate minima of the cepstrum.

By estimating the PSF-OOF and the PSF-MB, and then using the convoluted PSF of these two PSFs, we show that image sharpening with the Richardson-Lucy algorithm can detect almost the entire length of a crack on the surface of a concrete wall, though only half of its actual length could be detected in the degraded image [28]. In addition to applying cepstrum analysis to the sharpening of images captured by a camera mounted on a moving object by assuming that the amount and direction of movement of a UAV used for bridge inspection is one-dimensional for a short period of time, such as the shutter speed, this study confirmed that it is possible to narrow down the candidates for motion blur in cepstrum analysis, which has been difficult to narrow down the estimation in the past due to the susceptibility of the original image. By making it possible to efficiently sharpen blurred images using a general-purpose camera and a general-purpose PC, it is expected that bridge inspections using UAV will be realized as a solution to the current and future problem of a shortage of engineers in the inspection of more than 700,000 bridges in Japan.

The rest of the paper is organized as follows: Section 2 presents the Basic outline of the proposed method for the estimation of motion blur. Section 3 depicts the Experimental procedure and results. We prepared 4 different types of feature point and check the effect of these feature points on a cepstrum analysis and selection method of multiple feature points to narrow down the candidates of parameters of motion blur. Section 4 provides the Confirmation of the effectiveness of the proposed method. We estimated PSF-MB of the blurred image, captured by general-purpose camera, by using our proposed method and confirmed the sharpening effect of estimated PSF. Conclusion and future work suggestions are depicted in Section 5.

This paper is an extended version of [28] and we propose a method for specifying one PSF-MB candidate for each image and this method shows better sharpening result than the method in [28].

## 2. Basic Outline of the Proposed Method

The authors note that Δ(x,y) is an image that has only one non-zero brightness pixel at the origin and zero brightness pixels except origin, and the result of convolution of the PSF h(x,y) with the original image is h(x,y) itself [27]. The authors proposed that PSF for deconvolution of a degraded image captured by a camera on a moving object should be a convoluted PSF obtained by convolution of PSF-OOF and PSF-MB [28]. Since the PSF-OOF originates from the camera itself, it can be applied to multiple images acquired under the same condition by checking them at the beginning of the shooting. Therefore, it is possible to estimate PSF-OOF by preparing an artificial point texture (size and shape are known) in advance. In contrast, PSF-MB needs to be estimated for each image. As shown in Introduction, in this study, $C_{f}$ cannot be known, so it may be difficult to accurately determine the minimum value of $C_{h}$, such as when $C_{f}$ has a large value around the minimum value, simply by obtaining the cepstrum of the entire degraded image [20]. When a cepstrum analysis is performed on the linear blur degraded image, as shown in (Figure 1), a pixel that gives the minimum value to the cepstrum (hereinafter referred to as “pixel of minimum”) appears at the same position as the blur amount that is the cause of the degradation. Such a minimum appears at a position symmetrical to the origin [29]. In addition, it is known that at the same period (same direction of movement, integer multiple amount of movement), pixel of minimum (the absolute value of the minimum decreases due to decay) appears [22].

The following three points were confirmed, assuming an actual bridge inspection.

Confirmation of the influence of the feature points in the original image on the cepstrum;Our main target is a concrete structure such as bridges, whose surface includes not only concrete but also devices and accessories. Concrete may appear different in the image depending on material and subjects. Thus we selected homogeneous areas consisting of only simple textures and areas containing complex objects such as multiple different appendages as sub-images. We experimentally proved that we can grasp the blur amount accurately if we obtain a cepstrum of the sub-image with simple texture in the whole original image. The cepstrum of the sub-image with complex objects will not contribute to obtain blur amount.Examination of the selection method of multiple images to narrow down the number of the pixels that give a minimum cepstrum;When we select multiple sub-images suitable for accurately grasping the blur amount, we checked whether it is possible to narrow down the number of the candidates for pixels of minimum by taking the average of the cepstrum of selected sub-images. In addition, the requirements for selecting multiple images were organized, such as the required number and the characteristics of the images to be combined (e.g., nearby areas with the same characteristics or areas that are visually different due to differences in brightness or paint).Confirmation of the effectiveness of the proposed method;In order to confirm the effectiveness of the proposed method, we estimated the PSF-MB and confirmed the deconvolution effect by our convoluted PSF on the blurred image of actual concrete structure taken while moving the camera.

## 3. Experimental Procedure and Results

In order to verify that the estimation of the blur amount in a degraded image due to camera shaking works effectively, it is necessary to use a degraded image in which the blur amount is known in advance. Therefore, we prepared a degraded image where a filter which corresponds to a motion blur is convoluted. As the blur amount is known, it becomes possible to check if it could be estimated correctly as follows. We choose a concrete structure image (Figure 2) as an example [captured under the shooting condition shown in (Table 1)] and a filter giving the motion blur effect, which were made with MATLAB function [fspecial], set as shown in (Table 2) is convoluted to this image. We call this convoluted image a composite degraded image (hereinafter referred to as “com-degraded image”). Assuming the motion of a mobile object equipped with a camera, the amount of motion blur was set to about 10 pixels and the direction of motion blur was set to horizontal, vertical, or diagonal. Then the sub-images shown in Section 3.1 were extracted from the com-degraded image, and for these sub-images the amount of each blur was estimated with cepstrum analysis to check if it could be estimated correctly. To confirm the effect of this method, we add the experimental data of different amount of blur, five pixels and 15 pixels of two directions, horizontal and vertical.

### 3.1. Extraction of Sub-Images with Different Feature Points in the Original Image

In the original image taken with a fixed camera, small parts of image (121 × 121 pixels) are extracted from 4 areas (A to D) (Figure 2) with different feature points. The size of small parts of image (121 × 121 pixels) is decided as 12 × *n* (“*n*” is the amount of blur), described as in Section 3.2. The feature points were checked manually and the areas that we thought appropriate as feature points were extracted. In total 16 images, four images each, are extracted (hereinafter referred to as “sub-image”). Figure 3 shows one of the four sub-images extracted from area A to D respectively. The properties of the sub-images in area A to D are as follows. In the following, the numbers (1 to 4) in lowercase letters of the area names are used to distinguish the 16 sub-images (e.g., the four sub-images extracted from area A are represented by a1, a2, a3, and a4, respectively).

Area A: Isolated point-like texture as uniform feature points consisting of only simple shapes.Area B: Isolated point-like texture such as A, but in a different material or paint compared to area A.Area C: Texture with simple but non-uniform features such as dirt, cracks, etc.Area D: Uneven and complex texture with appendages of different shapes and materials.

### 3.2. Effect of Sub-Images on a Cepstrum Analysis

The cepstrum analysis was performed on a total of 16 sub-images shown in Section 3.3. From the results, we checked whether a pixel of minimum appeared at the place corresponding to the blur amount and at the next cycle (the same direction and twice the amount of movement as the blur amount) (hereinafter referred to as “next cycle”). The confirmation was done as follows (Figure 4).

With MATLAB function (imregionalmin), the pixels in each sub-image (121 × 121 pixels) with the smallest value compared to the nearest 8 pixels was identified as the pixel of minimum. These results were converted to a binary data, in which 1 was given to the pixel of minimum and 0 to the other pixels, and output them.The amount of movement *n* was 10 pixels or less, a range of 2 (*n* + 1) = 22 both vertically and horizontally from the center (totally 45 × 45 pixels) was assumed enough to check whether there was a pixel of minimum at the blur amount and the next cycle. Thus the central part (45 × 45 pixels) was extracted from the output binary data (121 × 121 pixels).For the extracted binary data (45 × 45 pixels), if 1(the pixel of minimum) appeared at the place of the blur amount and at the next cycle was checked. For each area (A to D), the amount of motion blur was determined to be estimable under the condition that 1 appeared at the place of the blur amount in all four sub-images. In this check, it was not possible to completely eliminate the effects of feature points contained in the original image itself, in addition to blur, which was the cause of degradation. Thus there were also pixels of minimum in addition to the place corresponding to the blur amount. Therefore, 1 appeared at the place different from the position that indicated the blur amount, and could not be eliminated. The solution to this problem is shown in Section 3.4. Even when 1 didn’t appear in the next cycle but 1 appeared at the same position as the blur amount, it was assumed that the amount of motion blur was determined to be estimable, in that case the reason that pixel of minimum did not appear in the next cycle is thought to be due to attenuation.

### 3.3. Estimation Results of Blur in Degraded Images

A cepstrum analysis was performed on the com-degraded image that is convolved with a filter with parameters as shown in (Table 2). Examples of cepstrum images obtained for each setting (amount and direction of movement) are as follows, one from each of four sub-images a1 to a4, b1 to b4, c1 to c4 and d1 to d4 extracted from area A to D respectively.

The blur amount (10 pixels, 0 degrees);As shown in (Table 3), all four sub-images extracted from area A and B have a pixel of minimum at the same position as the blur amount (Figure 5). From this, it is concluded that the blur amount can be estimated for area A and B.The blur amount (10 pixels, 90 degrees);As shown in (Table 4), all four sub-images extracted from area A, B, and C have a pixel of minimum at the same position as the blur amount (Figure 6). From this, it is concluded that the blur amount can be estimated for area A, B, and C.The blur amount (9 pixels, 26 degrees);As shown in (Table 5), all four sub-images extracted from area A and B have a pixel of minimum at the same position as the blur amount (Figure 7). From this, it is concluded that blur amount can be estimated for area A and B.The blur amount (7 pixels, 45 degrees);As shown in (Table 6), all four sub-images extracted from area A, B, and C have a pixel of minimum at the same position as the blur amount (Figure 8). From this, it is concluded that the blur amount can be estimated for area A, B, and C.The blur amount (5 pixels, 0 degrees, 90 degrees);As shown in (Table 7), for direction of horizontal and vertical, all four sub-images extracted from area A, B have a pixel of minimum at the same position as the blur amount. For direction of vertical, all four sub-images extracted from area C has a pixel of minimum at the same position as the blur amount. From this, it is concluded that the blur amount can be estimated for area A and B.The blur amount (15 pixels, 0 degrees, 90 degrees);As shown in (Table 8), for direction of horizontal and vertical, all four sub-images extracted from area A, B have a pixel of minimum at the same position as the blur amount. For direction of vertical, all four sub-images extracted from area C has a pixel of minimum at the same position as the blur amount. From this, it is concluded that the blur amount can be estimated for area A and B.

As shown in 1 through 4 in page parameter, for sub-images extracted from area A and B, where the feature points were uniformly isolated point-like textures consisting of only simple textures, for a movement amount of about 10, movement direction, horizontal, vertical, and diagonal movement, a pixel of minimum at the same position as the blur amount occurred in all conditions. For area C, which had simple but non-uniform stains and cracks as feature points, and area D, which had non-uniform and complex feature points such as appendages of different shapes and materials, there were sub-images for which no pixel of minimum appeared at the same position as the blur amount under several conditions. Therefore, it is confirmed that the blur amount can be estimated by cepstrum analysis using sub-images with uniform and simple shapes such as isolated point-like textures which are easily found on the surface of concrete structures.

### 3.4. Examination of the Selection Method of Multiple Sub-Images to Narrow down the Number of the Pixels of Minimum

#### 3.4.1. Procedure

A cepstrum was calculated for a total of eight sub-images extracted from area A and B, which had uniform isolated point-like textures consisting of only simple textures and were judged to be capable of estimating the blur amount in Section 3.3. Observed as images (45 × 45 = 2025 pixels), more than 100 pixels of minimum, including the pixel that represented the blur amount occurred. When multiple pixels of minimum, which are candidates for the blur amount, appear, it is necessary to obtain multiple PSF-MB that use each of them as the blur amount and identify the PSF-MB that gives the best sharpening effect as the blur amount. In consideration of the time and effort required for this process, it is desirable to reduce the number of the pixels of minimum.

Therefore, referring to Oyamada et al.’s study [22], we extracted several different sub-images from the com-degraded image and followed the following procedure to check whether it was possible to reduce the influence of $C_{f}$ by taking the average of each cepstrum of selected sub-images and narrowing down the number of the pixels of minimum (Figure 9).

Extract *n* (*n* is an integer) sub-images (45 × 45 pixels) from the com-degraded image and prepare a binary data (1 or 0) of the same size as the cepstrum image (45 × 45) obtained by cepstrum analysis.By summing the values (1 or 0) at each pixel (same coordinates) for *n* types of binary data (45 × 45), the value of each pixel (sum of *n* sub-images, 0 to *n*) is the number of the pixels of minimum at each pixel.For sum of the value of *n* types of binary data (45 × 45), by dividing the number of the occurrences of the pixel of minimum at each pixel by *n* (taking the average of *n* types of data), the numerical value at each pixel (45 × 45) becomes one of 0, 1/*n*, 2/*n*,..., *n*/*n* (=1).From the averaged values at each pixel, check the number of the pixels where all *n* pixels of minimum at the same location(*n*/*n* = 1) and their locations (Figure 9). Since the cepstrum image is the origin object and the pixel of minimum also occurs in the origin object, the two pixels of minimum that occur in the origin object are regarded as the same and counted as one (Figure 1).

Extracting four sub-images (*n* = 4) from area A and B respectively and checking them in the com-degraded images under the four conditions shown in (Table 2), it was possible to narrow down the number of the pixels of minimum (*n* = 4/4) to 6 or less in all conditions, as shown in (Table 9 and Table 10).

On the other hand, only the condition (iii) in area A (movement amount: 9 pixels; movement direction: 26 degrees) resulted in the occurrence of a pixel of minimum (*n* = 4/4) only at the same location as the blur amount and only one type of blur setting candidate could be identified. In the other conditions, there are multiple pixels of minimum (*n*= 4/4) that appear at different pixels from the pixel of the blur amount. This indicates that for feature points that show similar trend, the influence of the original image becomes similar. As a result, pixels of minimum from cepstrum of selected sub-images occur with similar tendency. Therefore, it is assumed that it would be difficult to narrow down the number of the blur amount candidates to one due to the occurrence of pixels of minimum in non-candidate locations originated from original image tendency.

We checked whether it was possible to narrow down the number of the blur amount candidates to one by combining sub-images extracted from area A and B that have different tendencies under the condition of *n* = 4. In the image of an actual concrete structure, there usually are areas with different brightness, such as sun and shade. Therefore, in the same subject as in (Figure 2), the dark areas of the same material as area A and B, which are extracted in Section 3.1 but in the shade are designated as area E and F (Figure 10 and Figure 11), respectively. For area E and F, with the same procedure as described in Section 3.3, we confirmed that the pixel of minimum occurred at the same location as the blur amount and that the blur amount could be estimated for all four settings in (Table 2), and then included them in the sub-image combinations. For the selection of sub-image combinations, we assumed that only two different conditions at most, such as “sun and shade at the same feature point” or “different feature points in sun or shade”, would be obtained on the surface of actual concrete structures in one picture, so the combinations among area A, B, E and F were set as follows.

(a)Sun and shade (light and dark) of the same component;(i)2 images of area A and E respectively, total 4 images(ii)2 images of area B and F respectively, total 4 images(b)Sun or shade of different components;(i)2 images of area A and B respectively, total 4 images(ii)2 images of area E and F respectively, total 4 images

**Figure 10 sensors-22-01635-f010:**
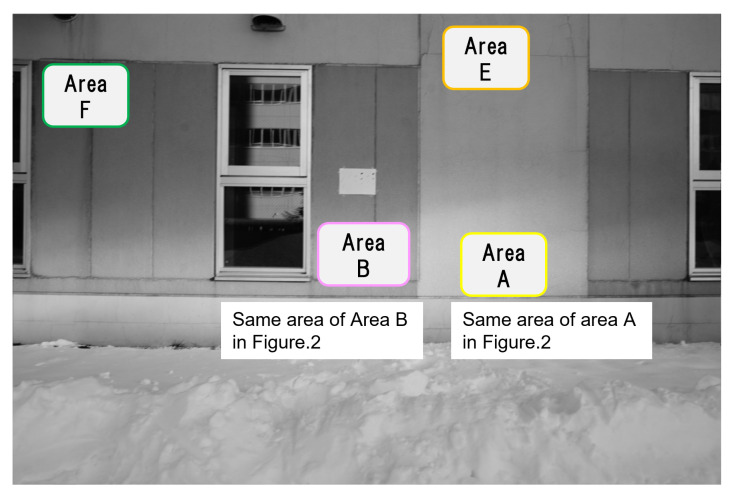
Areas with uniform and simple shapes as feature points captured by a fixed camera.

**Figure 11 sensors-22-01635-f011:**
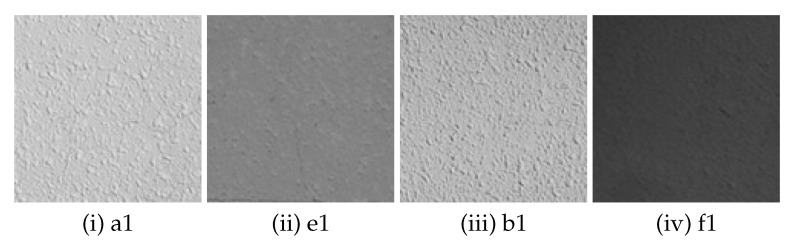
Examples of sub-images extracted from area A, B, E and F captured by a fixed camera.

#### 3.4.2. Results

Sun and shade (light and dark) of the same component (area A and E);As shown in (Table 11), the number of the blur amount candidates could be narrowed down to one in three out of four conditions in the combination of area A and E.Sun and shade (light and dark) of the same component (area B and F);As shown in (Table 12), the number of the blur amount candidates could be narrowed down to one in all four conditions in the combination of area B and F.Sun of different components (area A and B);As shown in (Table 13), the number of the blur amount candidates could not be narrowed down to one in the combination of area A and B in all four conditions.Shade of different components (area E and F);As shown in (Table 14), the number of the blur amount candidates could be narrowed down to one in one out of four candidates in the combination of area E and F in all four conditions.

**Table 11 sensors-22-01635-t011:** Sun and shade; The number of the pixels and their locations where multiple pixels of minimum appeared out of four sub-images out of area A and E.

The Blur Amount		The Number and Location of Multiple Pixels of Minimum Out of 4			
**Amount**	**Direction**	**4/4**	**3/4**	**2/4**	**Location 4/4 Pixels of Minimum Appear**
10	0	1	8	57	The blur amount only
10	90	3	17	91	The blur amount, 2 others
9	26	1	3	60	The blur amount only
7	45	2	9	55	The blur amount, next cycle

**Table 12 sensors-22-01635-t012:** Sun and shade; The number of the pixels and their locations where multiple pixels of minimum appeared out of four sub-images out of area B and F.

The Blur Amount		The Number and Location of Multiple Pixels of Minimum Out of 4			
**Amount**	**Direction**	**4/4**	**3/4**	**2/4**	**Location 4/4 Pixels of Minimum Appear**
10	0	1	2	56	The blur amount only
10	90	1	6	61	The blur amount only
9	26	1	9	72	The blur amount only
7	45	2	5	72	The blur amount, next cycle

**Table 13 sensors-22-01635-t013:** Sun of different components; The number of the pixels and their locations where multiple pixels of minimum appeared out of four sub-images out of area A and B.

The Blur Amount		The Number and Location of Multiple Pixels of Minimum Out of 4			
**Amount**	**Direction**	**4/4**	**3/4**	**2/4**	**Location 4/4 Pixels of Minimum Appear**
10	0	3	9	64	The blur amount, next cycle, other
10	90	4	4	62	The blur amount, next cycle, 2 others
9	26	3	5	60	The blur amount, 2 others
7	45	6	7	63	The blur amount, next cycle, 4 others

**Table 14 sensors-22-01635-t014:** Shade of different components; The number of the pixels and their locations where multiple pixels of minimum appeared out of four sub-images out of area E and F.

The Blur Amount		The Number and Location of Multiple Pixels of Minimum Out of 4			
**Amount**	**Direction**	**4/4**	**3/4**	**2/4**	**Location 4/4 Pixels of Minimum Appear**
10	0	1	6	59	The blur amount only
10	90	4	18	95	The blur amount, 3 others
9	26	2	4	63	The blur amount, other
7	45	3	7	64	The blur amount, next cycle, other

As shown in 1 and 2, under four different conditions, the combination of sun and shade (light and dark) (two types) of the same component was able to specify one blur amount in seven out of eight settings (four conditions x two types). On the other hand, as shown in 3 and 4, the blur amount could be narrowed down to one in only one setting out of eight settings for the combination of sun or shade of different components. Therefore, it is confirmed that it is possible to narrow down the number of the pixels of minimum as a candidate for the blur amount to a single location by combining two types of images with different brightness and taking the average of the cepstrum of selected sub-images even if they have similar feature points.

Next for the combination of images that have multiple blur amount candidates under the condition of *n* = 4, we increased *n* to 6 or more and take the average of them. For the combination of sun and shade (light and dark) of the same component, the combination of area A and E with the setting of movement amount 10 pixels and direction 90 degrees (only one setting that could not narrow down the number of the pixels of minimum to one under the condition of *n* = 4), it was possible to narrow down the number of the blur amount candidates to one.

For the combination of sun or shade (light or dark) of different components, there were seven settings that could not narrow down the number of the pixels of minimum to one under the condition of *n* = 4. For that seven settings, the blur amount could be identified in one setting for *n* = 6 and two for *n* = 8, and the number of the blur amount candidates could not be narrowed down to one in the remaining four settings even for *n* = 8 as shown in (Table 15 and Table 16). From the above, the requirements for feature points and combinations to reduce the influence of the original image in cepstrum analysis and to narrow down the number of the pixels of minimum as a candidate for the blur amount is that the combination of at least two images of each of two different brightnesses, for a total of at least four images. As a result, it is confirmed that from the result of cepstrum analysis of single sub-image (45 × 45 = 2025 pixels), there can be more than 100 candidates of the blur amount, with our proposed method, it can be narrowed down to a minimum of one candidate.

## 4. Confirmation of the Effectiveness of the Proposed Method

We confirmed the effectiveness of the proposed method by the following procedures.

Shooting blurred image;An image of the same concrete structure as shown in (Figure 2) was captured under the shooting condition shown in (Table 17). The camera for shooting was fixed on a tripod and moved by hand. This image (Figure 12) is the same as the degraded image used in [28].Estimation of PSF-MB;We estimated PSF-MB of the blurred image using the proposed method described in Section 3, and confirmed whether it could be possible to narrow down the number of the pixels of minimum. The details are shown in Section 4.1.Image Sharpening by Convoluted PSF;We used the method shown in [28] to sharpen the blurred image and checked the effect of the sharpening. The details are shown in Section 4.2 and Section 4.3 respectively.

**Table 17 sensors-22-01635-t017:** Shooting condition of a shaked camera.

Camera	SONY-α6000
Resolution	6 million (3008 × 2000 pixels)
Shutter speed	1/30 s
Focal length	16 mm
Shooting distance	2100 mm
Shaked camera	Camera is fixed on a tripod, and the image is taken while the tripod is moved
	(the amount of movement and the direction of movement are unknown).

### 4.1. Estimation of PSF-MB

From the blurred image (Figure 12), we extracted two sub-images each from area A and E, for a total of four images (Figure 13), and by averaging the results of the cepstrum analysis of them, we were able to narrow down the number of the pixels of minimum to one. Based on the results, the blur amount was estimated to be 5 pixels of movement amount and 11 degrees of movement direction. The blur amount obtained in this study was different from that estimated in [28] (6 pixels of movement and 18 degrees of direction). In [28], the spectra of four sub-images were obtained, summed, and the pixel with the smallest value of sum of four spectra was used as pixel of minimum equivalent to the blur amount. Therefore, since with the estimation method in [28] it was not confirmed that the pixel estimated as the blur amount gave a minimum value for each of four sub-images, the method proposed here is considered to be more reliable.

**Figure 12 sensors-22-01635-f012:**
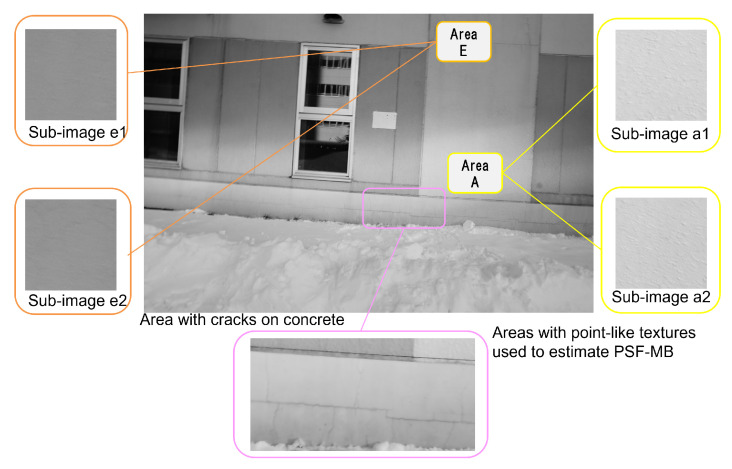
Blurred image captured by a shaked camera.

**Figure 13 sensors-22-01635-f013:**
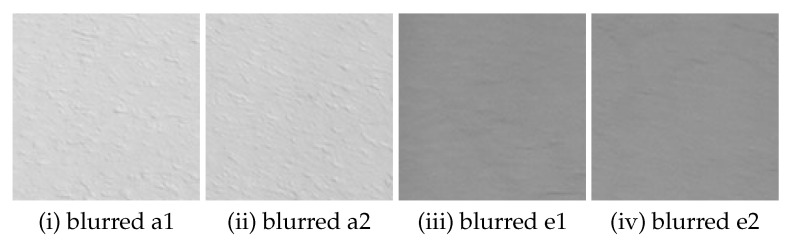
Examples of blurred sub-image from area A and E (121 × 121 pixels).

### 4.2. Sharpening by Convoluted PSF

PSF for deconvolution of blurred image captured by a camera on a moving object is obtained by convolution of two PSFs: PSF-OOF by camera system such as lens and PSF-MB by camera movement. In this paper the method shown in [28], convoluted PSF, was used to sharpen the blurred image. The details are as follows.

1.PSF-OOF estimation;Since the PSF-OOF is caused by the camera itself and can be applied to multiple images acquired under the same conditions by checking it at the beginning of the shooting, we estimated the PSF-OOF, $h_{1}\left(x,\;y\right)$, based on the target image [known size and shape, Black Circle (hereinafter represented by “BC”) and Black Square (hereinafter represented by “BS”) ] prepared in advance. As shown in [28], it is not possible to narrow down PSF-OOF candidates to one at this stage, but in the following, we will focus on BS size 2, B = 5, which had the highest sharpening effect among the 12 convoluted PSF candidates.2.Estimation of PSF-MB;We estimated PSF-MB of the blurred image using the proposed method described in Section 3, and confirmed whether it was possible to narrow down the number of the pixels of minimum.3.Estimation of convoluted PSF;The PSF-OOF, $h_{1}\left(x,\;y\right)$, was convolved with the PSF-MB, $h_{2}\left(x,\;y\right)$, movement amount 5, and movement direction 11 (degrees), and the convoluted PSF h(x,y) = $h_{2}\left(x,\;y\right)*h_{1}\left(x,\;y\right)$ which is the cause of the image degradation, was created by the following procedure (Figure 14).N is a number greater than B + 2L + 1 when the size of PSF-OOF is B and the amount of motion blur is L. In this case, N is set to 31 because B = 9 (maximum) and L = 5.(a)Prepare a filter of size N×N and luminance value P [luminance value of the background of the marker used to estimate PSF (out-of-focus) $h_{1}\left(x,\;y\right)$, set to 255 in this case].(b)Replace B×B (B = 5) from the center of the N×N filter with the PSF-OOF $h_{1}\left(x,\;y\right)$ calculated in (a) to create the N×N filter with PSF-OOF.(c)For N×N filter with PSF-OOF, N×N filter with convoluted PSF is created by convolving the blur amount [the amount of L (L = 5) and the direction of T (T = 11) estimated by cepstrum analysis (Figure 15)].(d)Extract F×F from the center of the image and normalize it to the convoluted PSF. Size F is origin object and odd number, so F = B + L or B + L + 1. Thus this case F = 11 because B = 5 and L = 5.4.Image sharpening by convoluted PSF;Since PSF is a kind of low-pass filter, high-frequency noise tends to be amplified in the deconvolution by Wiener filter. The deconvolution in this paper was performed in MATLAB using Richardson-Lucy’s (R-L) deconvolution algorithm [30,31], which has been highly evaluated for its correction effects and short processing time even for large images.

### 4.3. Sharpening Effect of Convoluted PSF

In blurred image (Figure 12), the area in which three cracks images ($C_{1}$ to $C_{3}$) (Figure 16: Captured by a fixed camera) observed was extracted as (Figure 17). The same area was detected from each sharpened image, and the crack detection rate (hereinafter represented by “CDR”) (average value of $C_{1}$ to $C_{3}$) was calculated by MATLAB and compared. The details are as follows.

Edge detection by differential operation using the Sobel gradient operator on the R-L deconvoluted image.Based on the differential value of the crack in the R-L deconvoluted image, the threshold of binarization was set so that the crack could be detected, this time the threshold was set to 51 (8 bit) and the binary image was skeletonized for the binarized imageOn the blurred image (Figure 17a) overlay the skeletonized image (Figure 17b), and compare the detected crack length with the crack length of the image acquired by a fixed camera (Figure 17c) to check CDR (crack length in the sharpened image/crack length in the image by a fixed camera). The crack length is defined as the length of a straight line connecting the two ends of a series of cracks in which at least one of the eight neighboring pixels is connected.Comparison of CDR in the sharpened image with CDR in the blurred image.

As shown in (Table 18), CDR was improved in all convoluted PSF compared with 0.51 in the blurred image (Figure 18b). It was confirmed that CDR of 0.87 was the highest in the image (Figure 19b) sharpened using convoluted PSF 11 × 11 pixels based on PSF (out-of-focus) estimated from BS size 2, B = 5. Comparing this result with that of [28], in [28] the highest CDR among the 12 convoluted PSF candidates was 0.85, and for 3 of 12 convoluted PSF, CDR was less than 0.51, which is the CDR of the blurred image. This indicates that our proposed method in this paper is more effective than the method in [28].

## 5. Conclusions and Future Work Suggestions

Images acquired by a camera mounted on a mobile object are subject to degradation, such as blurring of the camera focus due to shooting in dark areas or blurring of the acquired images due to shooting while a mobile robot is moving, which makes it difficult to check for cracks and other damages necessary for the maintenance of concrete structures. Therefore, in order to realize bridge inspection using a camera mounted on a mobile object such as a drone, it is desirable to estimate the PSF that causes blurring and to estimate the original image using the R-L algorithm or other methods. On the other hand, since PSF-MB, which is the cause of blur, differs from image to image, it is necessary to estimate it for each image after shooting. In this paper, we conducted experiments to confirm and examine the following two points using a general-purpose camera used in actual bridge inspections: (1) Influence on the cepstrum when the isolated point-like texture unique to concrete structures is used as a feature point. (2) Selection method of multiple images to narrow down the candidate minima of the cepstrum. In this study, isolated point-like textures that appear on the surface of concrete structures are used as uniform feature points consisting of only simple shapes. In addition, we propose a method to reduce the influence of the original image in a cepstrum analysis and to narrow down the candidates for PSF-MB, which is the cause of motion blur. Our proposed method can be applied to concrete structures in general as well as to subjects with similarly simple shaped feature points.

### Future Work Suggestion

Since this research was to confirm whether it is possible to perform sharpening in principle and the following points have not yet been confirmed, we would like to work on the following research in the future. The selection of feature points is manual, and we would like to be able to select them automatically by setting conditions. Extracting minima from multiple cepstrum analysis results is a manual process, and we would like to be able to extract minima automatically. We would like to estimate the uncertainty level related to the crack length or position of crack length by checking the effectiveness of this method on various structures and by taking statistics of the results.

Our goal is to realize bridge inspection by robotic technology using the method proposed here. In the future, we will confirm the effectiveness of our method on blurred images of concrete structures such as bridges acquired by robotic technology such as drones, and study how to select feature points (shape, brightness, etc.) with higher estimation accuracy.

We hope that our proposed method will reduce the time and effort required to sharpen degraded images and lead to the further use of robotic technologies such as drones for infrastructure maintenance.

## Figures and Tables

**Figure 1 sensors-22-01635-f001:**
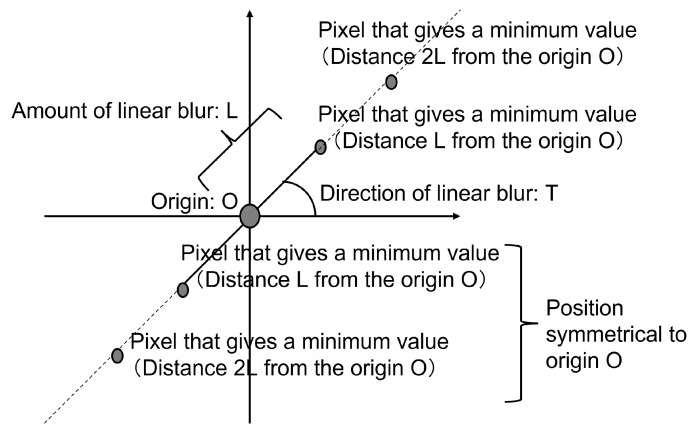
Schematic diagram of a cepstrum caused by linear blurring.

**Figure 2 sensors-22-01635-f002:**
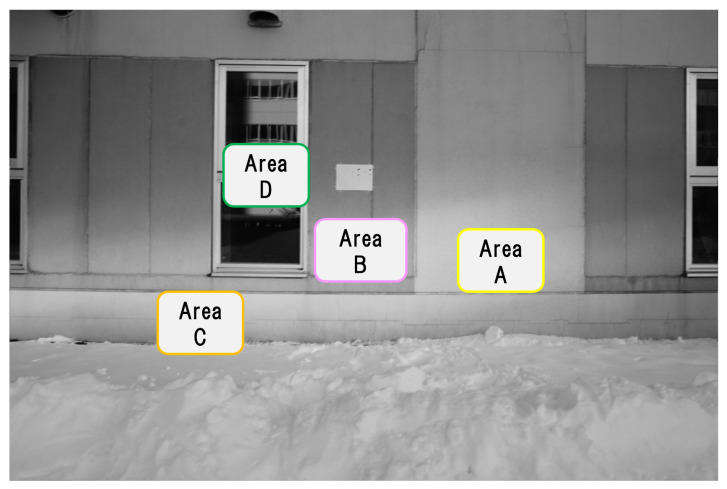
Original image captured by a fixed camera.

**Figure 3 sensors-22-01635-f003:**
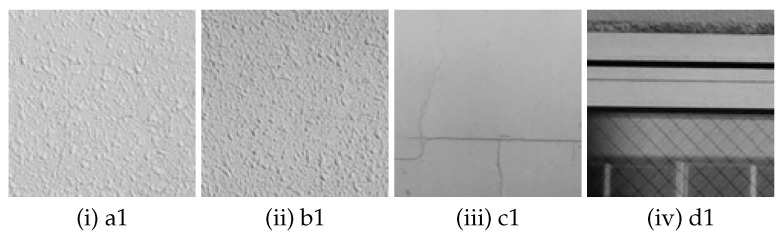
Examples of sub-image extracted from area A to D captured by a fixed camera.

**Figure 4 sensors-22-01635-f004:**
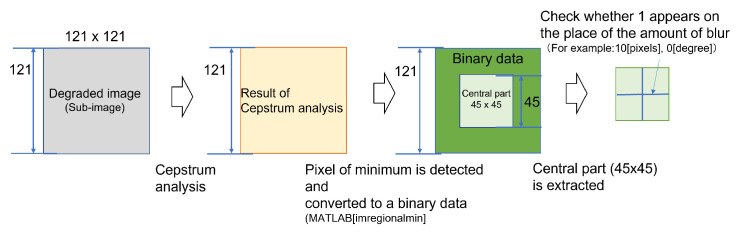
Procedure for confirming whether the blur amount can be estimated by cepstrum analysis.

**Figure 5 sensors-22-01635-f005:**
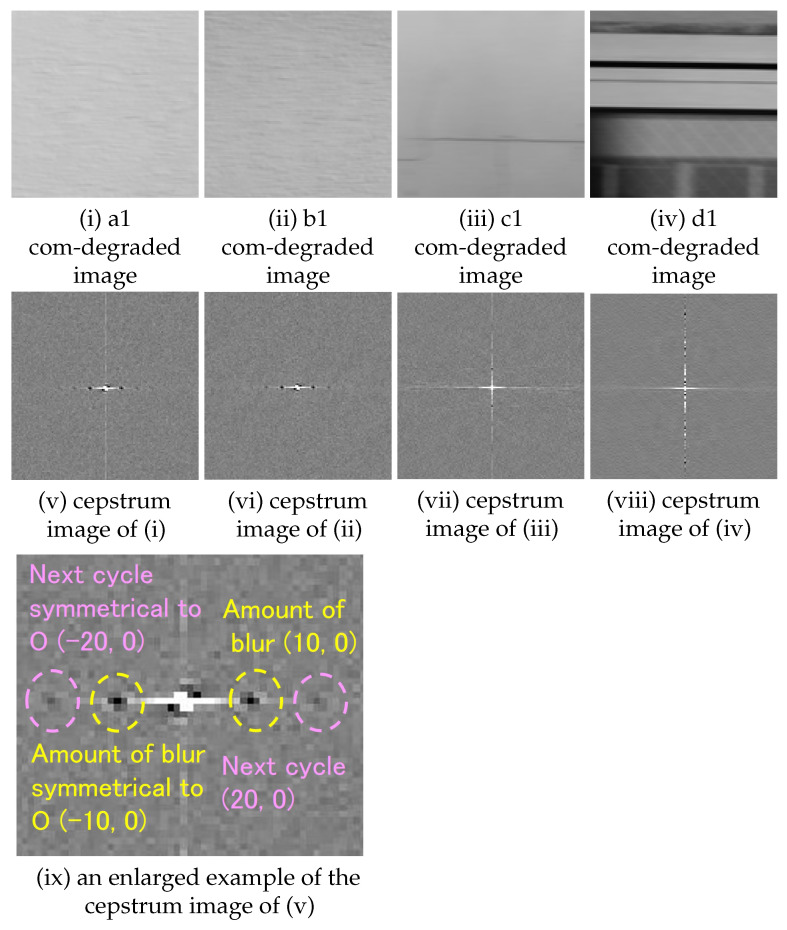
Examples of com-degraded image(10 pixels, 0 degrees) and cepstrum image from area A to D (121 × 121 pixels).

**Figure 6 sensors-22-01635-f006:**
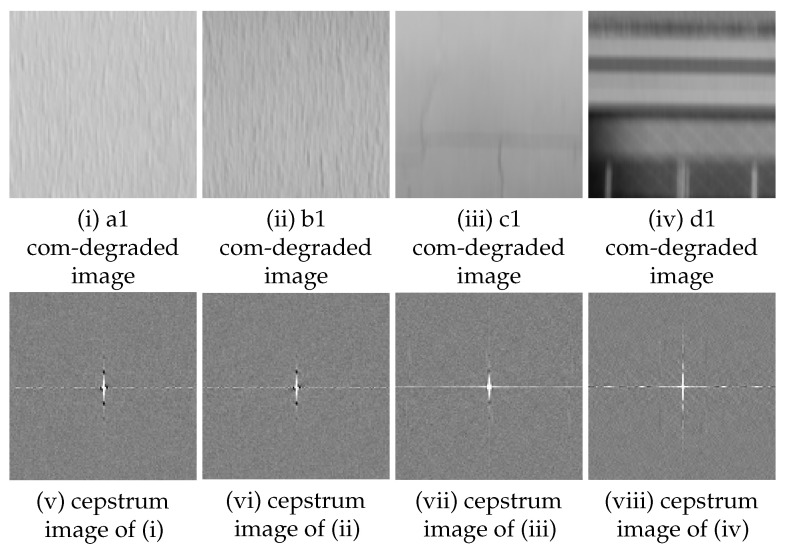
Examples of com-degraded image(10 pixels, 90 degrees) and cepstrum image from area A to D (121 × 121 pixels).

**Figure 7 sensors-22-01635-f007:**
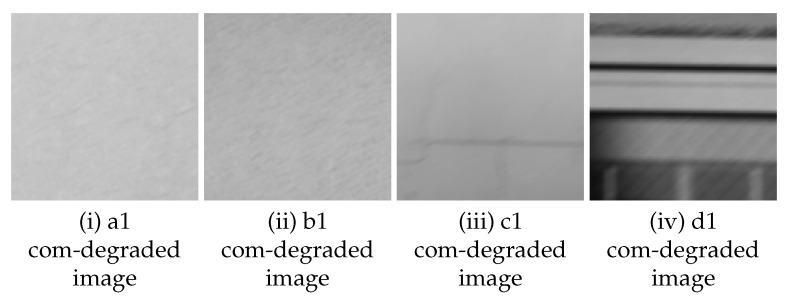
Examples of com-degraded image(9 pixels, 26 degrees) and cepstrum image from area A to D (121 × 121 pixels).

**Figure 8 sensors-22-01635-f008:**
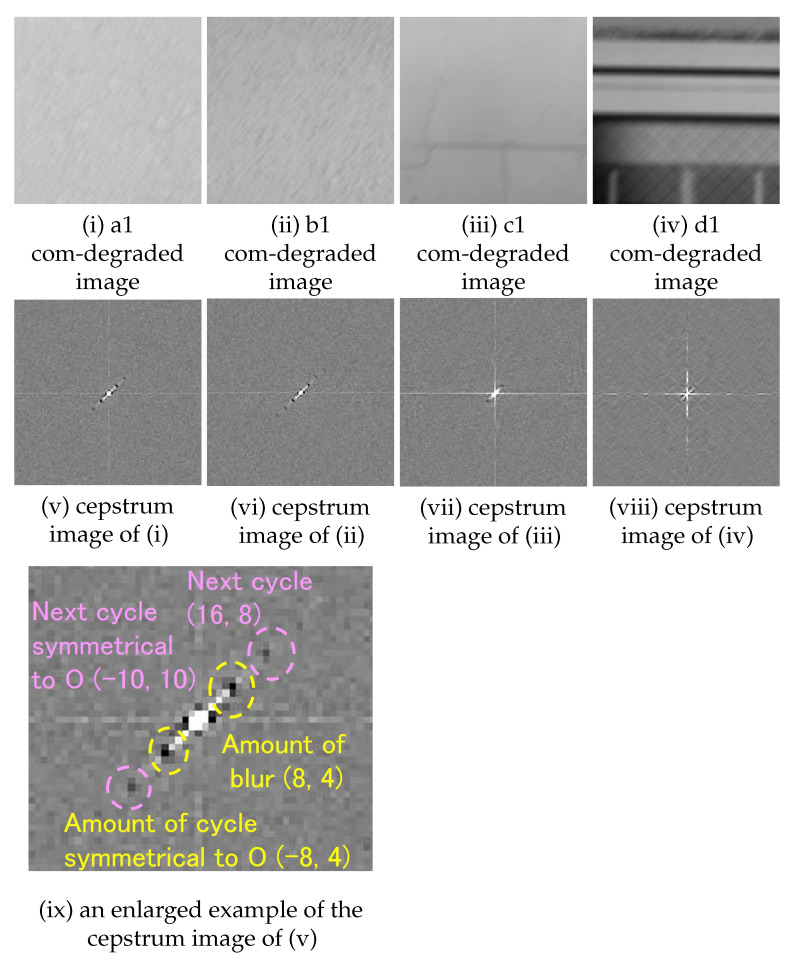
Examples of com-degraded image(7 pixels, 45 degrees) and cepstrum image from area A to D (121 × 121 pixels).

**Figure 9 sensors-22-01635-f009:**
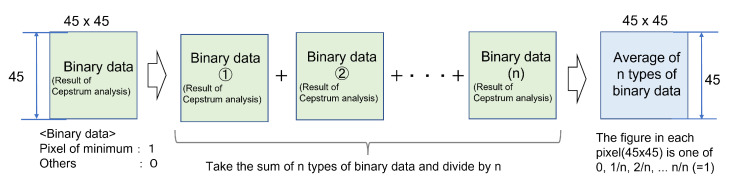
Procedure for checking the number and location of pixel of minimum in cepstrum analysis results.

**Figure 14 sensors-22-01635-f014:**
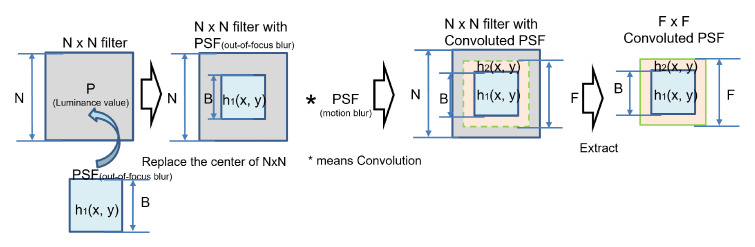
Creation of F×F convoluted PSF.

**Figure 15 sensors-22-01635-f015:**
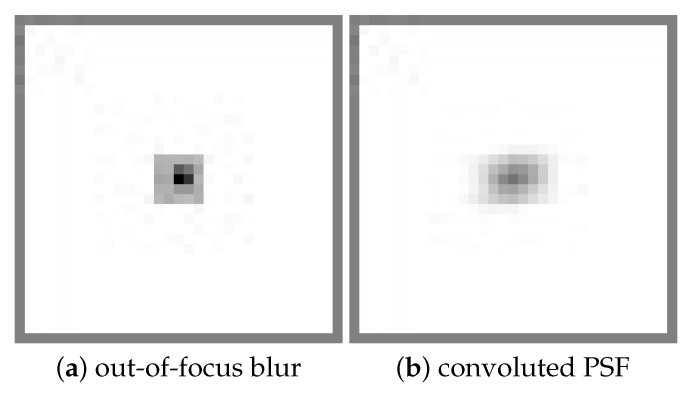
N×N filter with BS size2, B = 5 based estimated PSF.

**Figure 16 sensors-22-01635-f016:**
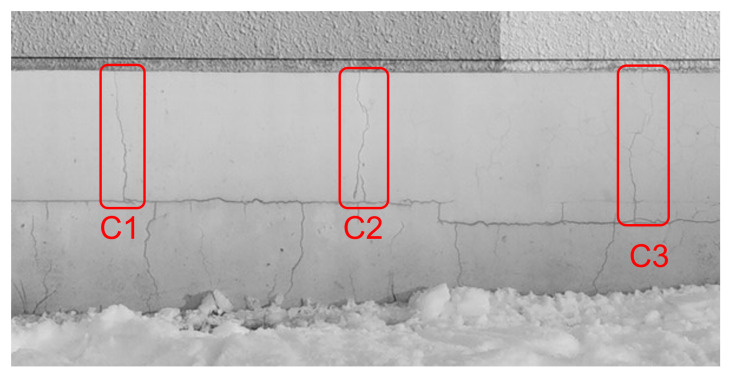
Three cracks captured by a fixed camera.

**Figure 17 sensors-22-01635-f017:**
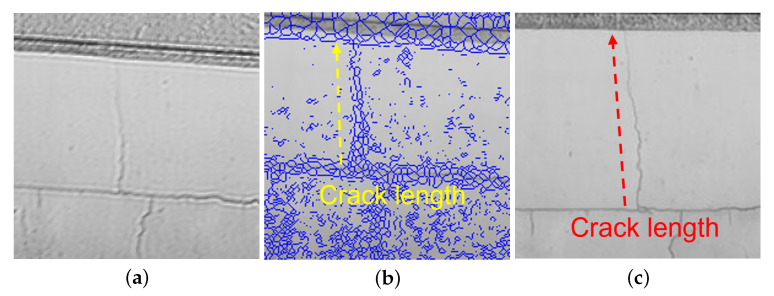
Crack image and crack length. (**a**) Sharpened image by deconvolution applying to motion blurred crack image; (**b**) Tracked crack (blue) superimposed on the image (**a**); (**c**) Crack image captured by a fixed camera.

**Figure 18 sensors-22-01635-f018:**
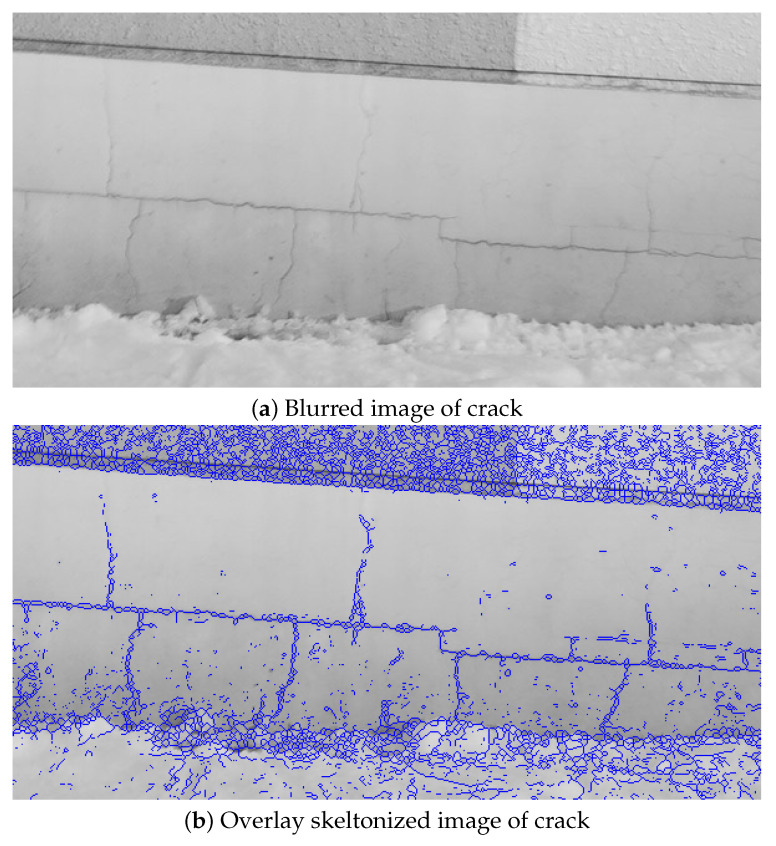
Blurred image and overlay skeltonized image of crack.

**Figure 19 sensors-22-01635-f019:**
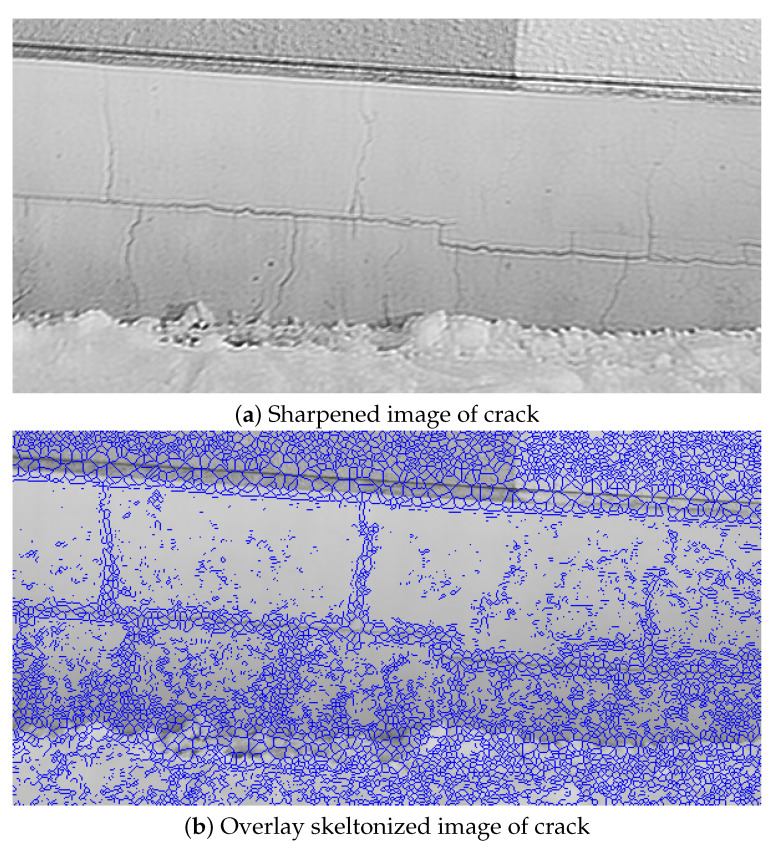
Sharpened image of crack deonvoluted with convoluted PSF (BS size 2, B = 5 based).

**Table 1 sensors-22-01635-t001:** Shooting condition of a fixed camera.

Camera	SONY-α6000
Resolution	6 million (3008 × 2000 pixels)
Shutter speed	1/30 s
Focal length	16 mm
Shooting distance	2100 mm
Fixed camera	Camera is fixed on a tripod.

**Table 2 sensors-22-01635-t002:** Settings of the blur amount.

Setting	Amount of Movement (pixels)	Direction of Movement (degrees)
i	10	0
ii	10	90
iii	9	26
iv	7	45

**Table 3 sensors-22-01635-t003:** The blur amount (10 pixels, 0 degrees).

Sub-Image		Pixel of Minimum Appears (OK) or None (×)		
**Area**	**No**	**Amount of Blur (10, 0)**	**Next Cycle (20, 0)**	**Estimation of the Blur Amount**
A	a1	OK	OK	Possible
A	a2	OK	OK	Possible
A	a3	OK	OK	Possible
A	a4	OK	OK	Possible
B	b1	OK	OK	Possible
B	b2	OK	OK	Possible
B	b3	OK	OK	Possible
B	b4	OK	OK	Possible
C	c1		×	Could not
C	c2	×	×	Could not
C	c3	OK	×	Possible
C	c4	×	×	Could not
D	d1	×	×	Could not
D	d2	×	×	Could not
D	d3	OK	×	Possible
D	d4	×	×	Could not

**Table 4 sensors-22-01635-t004:** The blur amount (10 pixels, 90 degrees).

Sub-Image		Pixel of Minimum Appears (OK) or None (×)		
**Area**	**No**	**Amount of Blur (0, 10)**	**Next Cycle (0, 20)**	**Estimation of the Blur Amount**
A	a1	OK	OK	Possible
A	a2	OK	OK	Possible
A	a3	OK	OK	Possible
A	a4	OK	OK	Possible
B	b1	OK	OK	Possible
B	b2	OK	OK	Possible
B	b3	OK	OK	Possible
B	b4	OK	OK	Possible
C	c1	OK	OK	Possible
C	c2	OK	×	Possible
C	c3	OK	×	Possible
C	c4	OK	OK	Possible
D	d1	×	OK	Could not
D	d2	×	×	Could not
D	d3	×	×	Could not
D	d4	×	×	Could not

**Table 5 sensors-22-01635-t005:** The blur amount (9 pixels, 26 degrees).

Sub-Image		Pixel of Minimum Appears (OK) or None (×)		
**Area**	**No**	**Amount of Blur (8, 4)**	**Next Cycle (16, 8)**	**Estimation of the Blur Amount**
A	a1	OK	×	Possible
A	a2	OK	×	Possible
A	a3	OK	OK	Possible
A	a4	OK	×	Possible
B	b1	OK	OK	Possible
B	b2	OK	×	Possible
B	b3	OK	OK	Possible
B	b4	OK	×	Possible
C	c1	OK	×	Possible
C	c2	OK	×	Possible
C	c3	×	×	Could not
C	c4	×	×	Could not
D	d1	OK	×	Possible
D	d2	OK	×	Possible
D	d3	OK	×	Possible
D	d4	×	×	Could not

**Table 6 sensors-22-01635-t006:** The blur amount (7 pixels, 45 degrees).

Sub-Image		Pixel of Minimum Appears (OK) or None (×)		
**Area**	**No**	**Amount of Blur (5, 5)**	**Next Cycle (10, 10)**	**Estimation of the Blur Amount**
A	a1	OK	OK	Possible
A	a2	OK	OK	Possible
A	a3	OK	OK	Possible
A	a4	OK	OK	Possible
B	b1	OK	OK	Possible
B	b2	OK	OK	Possible
B	b3	OK	OK	Possible
B	b4	OK	OK	Possible
C	c1	OK	OK	Possible
C	c2	OK	×	Possible
C	c3	OK	×	Possible
C	c4	OK	×	Possible
D	d1	OK	×	Possible
D	d2	OK	OK	Possible
D	d3	OK	×	Possible
D	d4	×	×	Could not

**Table 7 sensors-22-01635-t007:** The blur amount (5 pixels, 0 degrees and 90 degrees).

Sub-Image		Pixel of Minimum Appears (OK) or None (×)		
**Area**	**No**	**Amount of Blur (5, 0)**	**Amount of Blur (0, 5)**	**Estimation of the Blur Amount**
A	a1	OK	OK	Possible
A	a2	OK	OK	Possible
A	a3	OK	OK	Possible
A	a4	OK	OK	Possible
B	b1	OK	OK	Possible
B	b2	OK	OK	Possible
B	b3	OK	OK	Possible
B	b4	OK	OK	Possible
C	c1	×	OK	Possible (only vertical)
C	c2	×	OK	Possible (only vertical)
C	c3	OK	OK	Possible
C	c4	×	OK	Possible (only vertical)
D	d1	×	OK	Possible (only vertical)
D	d2	×	×	Could not
D	d3	×	×	Could not
D	d4	×	×	Could not

**Table 8 sensors-22-01635-t008:** The blur amount (15 pixels, 0 degrees and 90 degrees).

Sub-Image		Pixel of Minimum Appears (OK) or None (×)		
**Area**	**No**	**Amount of Blur (15, 0)**	**Amount of Blur (0, 15)**	**Estimation of the Blur Amount**
A	a1	OK	OK	Possible
A	a2	OK	OK	Possible
A	a3	OK	OK	Possible
A	a4	OK	OK	Possible
B	b1	OK	OK	Possible
B	b2	OK	OK	Possible
B	b3	OK	OK	Possible
B	b4	OK	OK	Possible
C	c1	×	OK	Possible (only vertical)
C	c2	×	OK	Possible (only vertical)
C	c3	OK	OK	Possible
C	c4	×	OK	Possible (only vertical)
D	d1	×	OK	Possible (only vertical)
D	d2	×	×	Could not
D	d3	×	×	Could not
D	d4	×	×	Could not

**Table 9 sensors-22-01635-t009:** The number of the pixels and their locations where multiple pixels of minimum appeared out of four sub-images out of area A.

The Blur Amount		The Number and Location of Multiple Pixels of Minimum out of 4			
**Amount**	**Direction**	**4/4**	**3/4**	**2/4**	**Location 4/4 Pixels of Minimum Appear**
10	0	3	7	62	The blur amount, next cycle, other
10	90	3	5	57	The blur amount, next cycle, other
9	26	1	6	64	The blur amount only
7	45	5	11	57	The blur amount, next cycle, 4 others

**Table 10 sensors-22-01635-t010:** The number of the pixels and their locations where multiple pixels of minimum appeared out of four sub-images out of area B.

The Blur Amount		The Number and Location of Multiple Pixels of Minimum Out of 4			
**Amount**	**Direction**	**4/4**	**3/4**	**2/4**	**Location 4/4 Pixels of Minimum Appear**
10	0	3	2	62	The blur amount, next cycle, other
10	90	4	6	47	The blur amount, next cycle, 2 others
9	26	4	3	73	The blur amount, 3 others
7	45	6	5	58	The blur amount, next cycle, 4 others

**Table 15 sensors-22-01635-t015:** The number of the pixels (n/n) and their locations where multiple pixels of minimum appeared out of sub-images out of area A and B (*n* = 4, 6, 8).

The Blur Amount		The Number and Location of Multiple Pixels of Minimum Out of *n*			
**Amount**	**Direction**	**8/8**	**6/6**	**4/4**	**Location 8/8 Pixels of Minimum Appear**
10	0	3	3	3	The blur amount, next cycle, other
10	90	3	4	4	The blur amount, next cycle, other
9	26	1	2	3	The blur amoun only
7	45	4	4	6	The blur amount, next cycle, 2 others

**Table 16 sensors-22-01635-t016:** The number of the pixels (n/n) and their locations where multiple pixels of minimum appeared out of sub-images out of area E and F (*n* = 4, 6, 8).

The Blur Amount		The Number and Location of Multiple Pixels of Minimum Out of *n*			
**Amount**	**Direction**	**8/8**	**6/6**	**4/4**	**Location 8/8 Pixels of Minimum Appear**
10	0	-	-	1	The blur amount specified (*n* = 4)
10	90	1	2	4	The blur amount only
9	26	-	1	2	The blur amount specified (*n* = 6)
7	45	2	2	3	The blur amount, other

**Table 18 sensors-22-01635-t018:** The crack detection ratio (CDR) of sharpened images by convoluted PSF.

Target	B	F	Avg ($C_{1}$–$C_{3}$) CDR	$C_{1}$DR	$C_{2}$DR	$C_{3}$DR
BC 1	5	11	0.80	0.74	0.95	0.70
BC 2	5	11	0.76	0.74	0.95	0.58
BS 1	5	11	0.82	0.72	0.94	0.79
BS 2	5	11	0.87	0.90	0.95	0.75
BC 1	7	13	0.83	0.77	0.90	0.81
BC 2	7	13	0.82	0.77	0.93	0.76
BS 1	7	13	0.70	0.75	0.90	0.46
BS 2	7	13	0.82	0.77	0.92	0.76
BC 1	9	15	0.58	0.81	0.49	0.44
BC 2	9	15	0.59	0.81	0.51	0.45
BS 1	9	15	0.58	0.81	0.50	0.44
BS 2	9	15	0.58	0.84	0.45	0.44

## Data Availability

The data or code presented in this study are available on request from the corresponding author.

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
