# Peer review of "Estimation and Sharpening of Blur in Degraded Images Captured by a Camera on a Moving Object"

_sensors, 2022, doi:10.3390/s22041635_

Round 1

Reviewer 1 Report

This manuscript proposed an image sharpening method to estimate concrete cracks based on PSF which is one of the common methods to sharpen blurred images. In this paper, the author estimated blur parameters by using the speckled pattern on the surface. The following points need to be considered for revision.  

1. Is the selection of roi (sub-image) manual? Are there any restrictions on roi size?
2. Please analyze or explain the different degrees of motion blur. Or the approximate motion blur range that this method is applicable to.
3. If reduce the exposure time, it is easy to get a clearer crack image, so please explain the necessity of the proposed method (will it get more other information?).
4. If possible please compare this method with other related methods (to show pros and cons of the proposed method).
5. Is there any future work related to the current result (to improve the result)?

Author Response

We appreciate your kind review. We answer your suggestions as follows.

The Line numbers in the rest of these answers indicate the Lines in the revised manuscript. The revised manuscript with highlighting the revised sentences is attached. 

Q1. Is the selection of roi (sub-image) manual? Are there any restrictions on roi size?

A1.We have added the sentense to L243-246.

Q2. Please analyze or explain the different degrees of motion blur. Or the approximate motion blur range that this method is applicable to.

A2.We agree with this comment. We have added the sentense to L238-240 and the experimental results of different amount of blur, 5 pixels and 15 pixels, to L309-320 and Table 7 and 8.

Q3. If reduce the exposure time, it is easy to get a clearer crack image, so please explain the necessity of the proposed method (will it get more other information?).

A3.Thank you for your suggestion. We have added the sentense to L44-46.

Q4. If possible please compare this method with other related methods (to show pros and cons of the proposed method).

A4.We agree with this comment. We have added to L128-142 the results of comparison of our proposed method with prior cases.

Q5. Is there any future work related to the current result (to improve the result)?

A5.Thank you for your suggestion. We have added  the sentense as future work suggestions to L 558-570 in Sec 5, and Sec 5 title is changed from "Conclusion" to "Conclusion and future work suggestions".

Reviewer 2 Report

(1) The abstract is not specific enough, and it is suggested to be perfect. For example, it does not clearly explain the specific means and methods used in the article. (2) The selection of keywords is not comprehensive. Although 3-5 keywords can be selected, it is recommended to select 5 keywords, and the selected keywords should be related to the main content of the article. (3) The article is not written in the format of sensor journal. For example, they are not given at the end of the article: Supplementary Materials, Author Contributions, Funding, Institutional Review Board Statement, etc. (4) The content of this article belongs to a long article. At the end of the Introduction part, the chapter structure of this article should be clearly explained to facilitate readers' reading. (5) The initial letter of the phrase in the figure should be capitalized. For example, "amount of linear blur" in Figure 1 should be "Amount of linear blur". Please revise this similar issue for all figures in this article. (6) The layout of figures and tables in the article should consider the format of the journal and the convenience of readers' reading through. Please check the full text in this article. It is recommended that Figure 17 be placed below Line 487. (7) There are some syntax errors that need to be modified as follows. Please check the full text in this article. 1. The initial "area" of Line 215, 217, 219 and 220 should be "Area". 2. The first letter in the table should be capitalized. For example, "setting" in Table 2 should be "Setting". Please check all tables in this article. 3. The font size in the figure shall be consistent with the font size in the figure and shall not be too large, such as Figure 17.

Author Response

We appreciate your kind review. We answer your suggestions as follows.

The Line numbers in the rest of these answers indicate the Lines in the revised manuscript. The revised manuscript with highlighting the revised sentences is attached. 

Q(1) The abstract is not specific enough, and it is suggested to be perfect. For example, it does not clearly explain the specific means and methods used in the article. 

A(1) Thank you for your suggestion. We have rewritten the abstract.

Q(2) The selection of keywords is not comprehensive. Although 3-5 keywords can be selected, it is recommended to select 5 keywords, and the selected keywords should be related to the main content of the article. 

A(2) Thank you for your suggestion. We have selected 5 keywords related to the main content of this article.

Q(3) The article is not written in the format of sensor journal. For example, they are not given at the end of the article: Supplementary Materials, Author Contributions, Funding, Institutional Review Board Statement, etc. 

A(3) Thank you for your suggestion. We have added Author Contributions and oters at the end of the article.

Q(4) The content of this article belongs to a long article. At the end of the Introduction part, the chapter structure of this article should be clearly explained to facilitate readers' reading. 

A(4) We agree with this comment. We have added the outline of this paper to L168-176.

Q(5) The initial letter of the phrase in the figure should be capitalized. For example, "amount of linear blur" in Figure 1 should be "Amount of linear blur". Please revise this similar issue for all figures in this article. 

A(5) Thank you for your suggestion. We have checked all figures in this article and modified some errors.

Q(6) The layout of figures and tables in the article should consider the format of the journal and the convenience of readers' reading through. Please check the full text in this article. It is recommended that Figure 17 be placed below Line 487. 

A(6) Thank you for your suggestion. We have checked all figures in this article and rearranged the location of them.

Q(7) There are some syntax errors that need to be modified as follows. Please check the full text in this article. 1. The initial "area" of Line 215, 217, 219 and 220 should be "Area". 2. The first letter in the table should be capitalized. For example, "setting" in Table 2 should be "Setting". Please check all tables in this article. 3. The font size in the figure shall be consistent with the font size in the figure and shall not be too large, such as Figure 17.

A(7) Thank you for your suggestion. We have checked all tables in this article and modified some errors.

Reviewer 3 Report

This is a very sound study that proposes a new method of make use of degraded images that could bring forward the process of structural health monitoring by saving time and reducing  the costs. Few minor corrections need to be considered prior publication.

  • Introduction line 66: The authors state that there has been a lot of research and then provide just a single references. The authors may rephrase or add few more references to support this statement.
  • Tables 3, 4, 5, 6 the authors should define what NG means.
  • Could an uncertainty level related to the crack length or position of crack length be estimated?
  • Although the conclusions section is comprehensive the authors may rewrite it highlighting the main findings and providing the guidelines for applying the method. Maybe a bulleting structure would be more helpful.

Author Response

We appreciate your kind review. We answer your suggestions as follows.

The Line numbers in the rest of these answers indicate the Lines in the revised manuscript. The revised manuscript with highlighting the revised sentences is attached. 

Q1. Introduction line 66: The authors state that there has been a lot of research and then provide just a single references. The authors may rephrase or add few more references to support this statement.

A1. We agree with this comment. We have added some references at L83.

Q2. Tables 3, 4, 5, 6 the authors should define what NG means.

A2. Thank you for your suggestion. We have changed "NG" to "Couldnot" 

Q3. Could an uncertainty level related to the crack length or position of crack length be estimated?

A3. Thank you for your suggestion. Since this research is to confirm whether it is possible to perform sharpening in principle. The uncertainty level related to the crack length or position of crack length is outside the focus of this research and is difficult to address at this time. In sec 5, we have added the sentence as future work suggestion to L564-566. 

Q4. Although the conclusions section is comprehensive the authors may rewrite it highlighting the main findings and providing the guidelines for applying the method. Maybe a bulleting structure would be more helpful.

A4. We agree with this comment. We have added the sentense to L547-551  and L555-557.

Round 2

Reviewer 1 Report

The authors answered all the questions and made the detailed revisions.